# Predicted Role of Mitsugumin 23 in Skeletal and Cardiac Muscle

**DOI:** 10.3390/cells14241958

**Published:** 2025-12-10

**Authors:** Daiki Watanabe, Richard C. Edel, Miyuki Nishi, Hiroshi Takeshima, Samantha J. Pitt

**Affiliations:** 1Graduate School of Sport and Health Sciences, Osaka University of Health and Sport Sciences, Osaka 590-0496, Japan; 2School of Medicine, University of St Andrews, St Andrews KY16 9TF, UK; 3Graduate School of Pharmaceutical Sciences, Kyoto University, Kyoto 606-8501, Japan

**Keywords:** Mitsugumin23/*TMEM109*, sarcoplasmic reticulum, calcium regulation

## Abstract

Mitsugumin 23 (MG23) is a transmembrane protein expressed in the nuclear membrane and endo/sarcoplasmic reticulum (ER/SR) of various tissues, including skeletal and cardiac muscle. MG23 is a non-selective cation channel that has been implicated in the leakage of calcium ions (Ca^2+^) under diverse pathophysiological conditions. SR Ca^2+^ leak is considered to be a contributing factor of skeletal muscle weakness and is also implicated in the progression of heart failure. The absence of MG23 has been reported to alleviate negative outcomes associated with SR Ca^2+^ leak. Targeting MG23 could represent a new therapeutic strategy against muscle disorders. This review discusses the potential role of MG23 in skeletal and cardiac muscle, and highlights MG23 as both a regulator of basal SR Ca^2+^-handling and a mediator of pathophysiological remodelling in muscle.

## 1. Introduction

In both skeletal and cardiac tissue, muscle contraction is driven by a process termed excitation–contraction coupling (ECC). ECC was first coined in the early 1950s [1] and is a physiological process linking the excitation of muscles by the nervous system to mechanical force to initiate contraction. In both skeletal and cardiac muscle, action potential propagation is followed by the release of Ca^2+^ from the sarcoplasmic reticulum (SR). Interaction between elevated cytosolic [Ca^2+^] and the myofilament protein troponin-C triggers a series of conformational changes in the troponin–tropomyosin complex. Resultant exposure of the myosin-binding site on actin allows for cross-bridge formation and subsequent muscle contractions to occur. In skeletal muscle, the type-1 ryanodine receptor (RyR1; SR Ca^2+^ release channel) opens by direct physical interaction with the dihydropyridine receptor (DHPR) within the transverse tubular (T-tubular) system [2]. In cardiac muscle, a localized increase in [Ca^2+^] mediated through DHPR opening is sufficient to directly activate the type-2 ryanodine receptor (RyR2). RyR2 channel activation is responsible for subsequent amplification of the cytosolic [Ca^2+^] in a process termed Ca^2+^-induced Ca^2+^ release (CICR; [3]).

Intracellular Ca^2+^ regulation by the SR plays a crucial role not only in muscle contraction but also muscle adaptation under pathophysiological conditions. Specifically, the leakage of Ca^2+^ from the SR is known to occur under various situations that modulate muscle function. For instance, in skeletal muscle, excessive Ca^2+^ leak from the SR activates Ca^2+^-dependent proteases such as calpain to enhance reactive oxygen species (ROS) production, which can lead to muscle weakness frequently observed in aged and diseased muscles [4,5,6]. In contrast, exercise-induced SR Ca^2+^ leak may trigger mitochondrial biogenesis [7]. In cardiac muscle, basal SR Ca^2+^ leak is thought to play an important physiological role in the subtle sensitization of RyR2 for activation through CICR [8] and cardiac protection against spontaneous Ca^2+^ release caused by SR-store Ca^2+^ overload [9]. However, SR Ca^2+^ leak can become pathogenic. At the molecular level, the uncontrolled leakage of Ca^2+^ from the SR store is a hallmark of many cardiovascular diseases including heart failure [10,11,12,13]. Therefore, SR Ca^2+^ leak exerts biphasic effects on muscle function: optimal Ca^2+^ leak offers positive effects, while excessive Ca^2+^ leak impairs muscle function.

Many studies have focused on the molecular mechanism of SR Ca^2+^ leak, including the identification of SR Ca^2+^-leak channels. RyR has been shown to act as a SR Ca^2+^-leak channel under various pathophysiological conditions in both skeletal [14,15,16] and cardiac [17,18] muscles. Although limited, recent evidence suggests that in addition to RyR-mediated SR Ca^2+^ leak, Mitsugumin 23 (MG23) may contribute to the leakage of Ca^2+^ across the SR [19,20,21]. In this review, we discuss the potential role of MG23 in cardiac and skeletal muscle based on the biological behaviour of MG23.

## 2. Regulation of Intracellular Ca^2+^ Homeostasis in Muscle

Intracellular Ca^2+^ is distributed in cytoplasmic and various organelles including endoplasmic reticulum (ER)/SR, mitochondria, T-tubular system, lysosomes, nuclei, and the Golgi apparatus. In skeletal muscle at rest, the majority of Ca^2+^ is distributed in the SR (~1.2 mM), and a small amount of Ca^2+^ is distributed within the mitochondrial and cytoplasmic region (Ca^2+^ concentrations are ~200 μM in mitochondria and ~50 nM in cytoplasm, respectively [15]). It has been demonstrated that intracellular Ca^2+^ is redistributed by the enhancement of SR Ca^2+^ leak [22,23]. With mitochondria being implicated in Ca^2+^ buffering [15,24], a growing number of studies have shown that the interaction between mitochondria and SR modulates muscle adaptations [7,25], muscle injury [26,27], and muscle diseases [18,28,29]. Furthermore, recent studies proposed a negative spiral in which SR Ca^2+^ leak increases mitochondrial Ca^2+^, which enhances mitochondrial ROS production, leading to further SR Ca^2+^ leak due to the oxidative modification of related channels such as RyR [30]. In summary, intracellular Ca^2+^ in muscle is mainly regulated by SR, and mitochondria likely contribute to intracellular Ca^2+^ homeostasis under conditions where SR Ca^2+^ leak is enhanced. Moreover, the Ca^2+^ redistribution from the SR to the mitochondria may play a significant role in determining the pathophysiological status of muscle.

## 3. Molecular Basis of Mitsugumin 23

Mitsugumin 23 (UniProt reference: Q9BVC6 for human and Q3UBX0 for mouse), encoded by *TMEM109* gene, was named by Japanese researchers based on its location within the muscle cell. The T-tubular system is an extension of the sarcolemma and forms close functional contact with the SR, allowing for rapid signal transduction between them. In cardiac muscle, electron microscopy images in transverse sections have shown that one T-tubule membrane and one SR cisterna form a close association, whereas in skeletal muscle, one T-tubule is sandwiched between two SR cisternae. These structures of SR(s) and T-tubules are called dyads and triads, respectively, due to their morphological features. In Japanese, a triad is called “Mitsugumi Kouzou” and many proteins discovered in the triad by Japanese researchers are named “Mitsugumin”, with the relative molecular weight added thereafter. In this context, many Mitsugumin proteins have been identified, including Mitsugumin53/*TRIM72*, Mitsugumin29/*SYPL2*, and Mitsugumin56/*HHATL*, as well as MG23/*TMEM109*. Nishi et al. (1998) first identified MG23 and characterized various biological aspects, including its expression in a wide variety of cell types and its structural characteristics [31]. Here, we describe the molecular basis of MG23 based on studies by Nishi et al. (1998) [31] and later researchers.

### 3.1. Expression

Mitsugumin proteins were first identified using monoclonal antibodies produced by the immunization of mice with heavy SR membrane fractions prepared from adult rabbit skeletal muscle. These SR extractions were enriched in triads, suggesting that Mitsugumin proteins are abundantly expressed within this region. Subsequent immunofluorescence microscopy and immunoblotting in skeletal muscle fibres revealed specific labelling to nuclear and cytoplasmic membranous components that were orientated transversely, suggestive of being the SR [31]. MG23 is expressed in a wide variety of cells, and its localization varies depending on the cell type. In striated muscle, MG23 localizes equivalently to the nucleus and SR, while in smooth muscle the relative MG23 expression in SR appears to be lower than that in the nucleus [31]. The defined localisation of MG23 may have significant physiological consequences on cellular function.

### 3.2. Structure

Nishi et al. (1998) proposed that MG23 possesses three transmembrane segments, with its amino terminus located in the luminal region and its carboxyl terminus in the cytoplasmic region [31]. This model was later supported by Venturi et al. (2011) [32]. Based on these findings, MG23 has been reported to consist of 210 amino acids, of which 59 residues (28%) are predicted to be in the cytoplasmic region, 65 residues (31%) in the transmembrane region, and 86 residues (41%) in the luminal region [31,32]. Interestingly, there is a significant sequence similarity between the cytoplasmic region of MG23 in the carboxyl-terminal region and the tail portion of the myosin heavy chain [31]. The tail portion of the myosin heavy chain is thought to play an important role in myosin filament assembly [33], suggesting that MG23 may interact with neighbouring MG23 or other cytoplasmic proteins.

Chemical cross-linking analysis of purified recombinant rabbit MG23 suggested that MG23 forms homo-oligomeric complexes up to a hexamer [32]. Three-dimensional reconstruction of electron microscopy images indicated that native MG23 exists either in a bowl-shaped molecule or a roughly crescent-shaped molecule [32] (Figure 1A). Although these structures are relatively low in resolution, it was suggested that the bowl-shaped molecule is 12 nm in height, 16 nm in side length, and 17 nm diagonally at the widest transmembrane region. In contrast, the crescent-shaped molecule is shaped as 14 nm in height, 11 nm in side length, and 6 nm in narrow side length. The exact number of MG23 subunits required for the formation of crescent- and bowl-shaped molecules remain uncertain. However, based on simulations derived from electron microscopy images, it has been proposed that the crescent-shaped molecule is composed of a hexamer of MG23, and the bowl-shaped molecule is assembled from six crescent-shaped molecules [32]. It remains unclear why MG23 forms these two different structures. Particle image analysis implies that the crescent-shaped subunits are unstably assembled into the bowl-shaped structure [32]. This raises the question, does the oligomerization of MG23 into the bowl-shaped structure lead to the destruction of ionic gradients across the ER to drive cell death and/or pathology?

### 3.3. Channel Properties

MG23 is considered to act as a non-selective cation channel [32,34]. Single-channel recordings of MG23 from rabbit skeletal muscle indicated voltage-regulated increases in K^+^ and Ca^2+^ currents [32], where the activity of MG23 was greater at negative compared to positive holding potentials. In a cellular environment, this inherent voltage dependence was proposed to render MG23 more open during refilling of the ER/SR stores or just before a Ca^2+^ release event [32]. The relative permeability of MG23 to Ca^2+^ and K^+^ is reported to be equal. Using the Fatt–Ginsborg equation to calculate the relative permeability of MG23 following the construction of current–voltage relationships under bi-ionic conditions, the PCa^2+^/PK^+^ was reported to be 1:1. Given the location of MG23 and the large Ca^2+^ gradient across the SR membrane, it was suggested that MG23 may play a role alongside RyR2 in shaping intracellular Ca^2+^-dynamics [20,32]. The permeability of MG23 to other monovalent and divalent cations has not been reported. Since MG23 is relatively non-selective, small changes in ionic composition across the SR together with channel expression changes may result in large changes to the current flux through this channel. Interestingly, single-channel MG23 recordings were characterized by two distinct features (Figure 1B): (1) brief and (2) coordinated gating. These events are likely to be explained by single-channel openings from a single functional unit and synchronized gating resulting from the co-ordinated opening of multiple channels. There appears to be no tendency for a preferred number of MG23 channels to gate together in synchrony. The unusual gating features might therefore reflect the structural transitions of the crescent-shaped subunit into the bowl-shaped assembly. Ionic conduction may therefore be reflected by the dynamic conformational changes and the oligomeric state of the protein. This poses the question, does the transient assembly and/or disassembly of the bowl-shaped particle observed from electron microscopy images underlie the sporadic activity of multiple MG23 channels opening in synchrony?

### 3.4. Potential Regulators

Regarding the regulation of MG23, there are currently no pharmacological tools that can selectively inhibit or modulate MG23 activity, which has limited studies of the role of this channel. Below, we discuss known and potential regulators of MG23.

#### 3.4.1. Zinc

We do know that the gating of MG23 prepared from cardiac tissue is regulated by Zn^2+^ [20] (Figure 2). It is likely that MG23 in skeletal muscle is also regulated by Zn^2+^, as it is coded by the same gene and, to date, there is no evidence to suggest that MG23 is expressed as a different isoform in different tissue. Cardiomyocytes contain a small but measurable pool of free Zn^2+^ in the cytosol, which is reported to be ~100 pM [35]. This basal level of Zn^2+^ is transiently altered during cardiac ECC as result of both the influx of Zn^2+^ into the cell through L-type Ca^2+^ channels and the release of Zn^2+^ from intracellular stores including the ER/SR [36]. Disrupted intracellular Zn^2+^ homeostasis has long been associated with the reduced cardiac contractility observed in heart failure [37,38,39]. Increasing the Zn^2+^-concentration gradient across the plasma membrane by application of high levels (10 μM to 20 mM) of extracellular Zn^2+^ was demonstrated to reduce isolated cardiomyocyte contractility as a result of SR-Ca^2+^ store depletion, suggesting an intimate relationship between raised intracellular Zn^2+^ and SR Ca^2+^ release [40,41]. Indeed, studies suggest that alterations in intracellular Ca^2+^ homeostasis and intracellular Zn^2+^ levels are more than coincidental and that they may exist in a synergistic relationship [36,42]. Using live-cell imaging approaches, elevations in intracellular Zn^2+^ were demonstrated to be in close proximity to the SR [39]. Zn^2+^ modulation of the SR Ca^2+^ load may therefore play an important role in cardiac contractility. Turan and colleagues have extensively investigated the proposed relationship between elevated intracellular Zn^2+^ levels, SR Ca^2+^ release, and reduced cardiac contractility. They showed that redox-dependent signalling pathways can elevate intracellular Zn^2+^ levels by the release of Zn^2+^ from SR stores, and oxidation-induced release of Zn^2+^ from Zn^2+^-binding proteins such as metallothioneins. These mechanisms result in a 30-fold elevation in the cytoplasmic Zn^2+^ level, resulting in myocardial damage [43,44,45,46].

Using cardiac SR vesicles isolated from sheep to study single-channel properties under voltage-clamp conditions, our previous work demonstrated that elevating the cytosolic [Zn^2+^] from 100 pM to 1 nM regulates the gating of both RyR2 and MG23 [20,47]. Furthermore, marked alterations in the expression of Zn^2+^ transporters have been observed as part of a compensatory response to rising intracellular Zn^2+^ levels in a pressure-overload (TAC) rat model, H9C2 cellular doxorubicin-induced heart-failure model, and in tissue from failing human hearts [21,48,49]. Elevated Zn^2+^ concentrations in these settings correlate closely with increased ER-stress markers, indicating that dysregulated zinc homeostasis contributes directly to ER-stress signalling during cardiac remodelling and failing hearts [50]. This suggests that Ca^2+^ homeostasis is intimately related to intracellular Zn^2+^ levels and suggests that physiological levels of Zn^2+^ are essential in fine-tuning the release of Ca^2+^ from the SR during cardiac ECC. Zn^2+^-induced SR Ca^2+^ leak through the modulation of both MG23 and RyR2 functions may therefore be a key mechanism in the progression of cardiac dysfunction in the failing heart.

#### 3.4.2. Oxidative Stress

Our preliminary observation suggests that the complex formation of MG23 from rabbit skeletal muscle is promoted by hydrogen peroxide, a long-living ROS. Because oligomerization can enhance the ion permeability of the MG23 channel, it is possible that ROS promotes its ion conduction. ROS typically react with cysteine residues, leading to disulphide formation and subsequent conformational changes in the protein. Interestingly, human MG23 does not have any cysteine residues, but oxidation may regulate the channel indirectly through altered phosphorylation and/or altered interactions with regulatory proteins. Oxidative stress has been linked to increased SR Ca^2+^ leak during myocardial infarction, heart failure [51]. Therefore, it may link to MG23-mediated Ca^2+^ leak under pathophysiological conditions.

#### 3.4.3. Phosphorylation

Although currently there is no experimental evidence to suggest that MG23 can be modulated by kinase activity, using the predictive software PhosphoSite (v6.8.2; https://www.phosphosite.org/), there are two putative phosphorylation sites on MG23, one of which is located within the cytoplasmic tail. Phosphorylation may therefore be a key regulator of MG23 channel function, potentially resulting in altered Ca^2+^ cycling across the SR. This is especially relevant given that heart failure is associated with widespread remodelling of kinase and phosphatase activity (for a review, see Ahmed et al. (2025) [52]).

## 4. Role of Mitsugumin 23 in Skeletal Muscle

Body movement is regulated by skeletal muscle contraction, and muscle contractile force is therefore a key determinant of physical performance. SR Ca^2+^ release plays a central role in force production, as muscle contraction is evoked by ECC. In skeletal muscle, SR Ca^2+^ release is triggered by the direct interaction of DHPR with RyR1. Thus, SR Ca^2+^ release is dependent on the voltage sensing of DHPR, the coupling between DHPR and RyR1, and the open probability of RyR1. SR Ca^2+^ content also plays a crucial role in determining SR Ca^2+^ release and thereby force production, because RyR opening is partially regulated by the luminal free-Ca^2+^ concentration [53]. When the SR Ca^2+^ content is below the normal physiological level, the amount of Ca^2+^ released in response to an action potential increases linearly with increased SR Ca^2+^ content [54]. However, once the SR Ca^2+^ content exceeds this level, the relationship plateaus [54]. This suggests that a decrease in SR Ca^2+^ content below the normal physiological range can impair force production, while an increase above this level has little or no additional effect on force production. This is important when we consider that the pathological leakage of Ca^2+^ from SR will result in reduced store content.

SR Ca^2+^ content under resting conditions is determined by the balance between SR Ca^2+^ leak and Ca^2+^ uptake, and the amount of SR Ca^2+^-binding proteins such as calsequestrin. Of these, SR Ca^2+^ leak is reported to be altered under pathophysiological conditions. RyR is known to act as SR Ca^2+^-leak channel, and RyR-mediated Ca^2+^ leak is regulated by various factors such as fragmentation [55] and phosphorylation [56]. In addition to RyR, SR Ca^2+^ leak via SR Ca^2+^-ATPase has been proposed to occur if the temperature is elevated, possibly due to the high production of superoxide from the mitochondria [57,58]. However, under certain conditions, SR Ca^2+^ leak is not fully blocked even when RyR and SR Ca^2+^-ATPase inhibitors are applied [14], suggesting the existence of other SR Ca^2+^-leak channels.

Our recent work suggests that MG23 may contribute to SR Ca^2+^ leak in skeletal muscle [19]. Consistent with this idea, our latest data shows that endogenous SR Ca^2+^ content is slightly increased in type I and type II fibres lacking MG23 (Figure 3) [19]. Furthermore, force production remained unchanged in fast- and slow-twitch muscles lacking MG23. This is consistent with the findings of Posterino and Lamb (2003) [54], who demonstrated that, in skeletal muscle, force production is unaltered when the SR Ca^2+^ content rises above physiological levels and is only impaired when levels fall below this value.

Despite increased SR Ca^2+^ content, *Mg23* knockout mice did not exhibit a marked difference in SR Ca^2+^ leak [19]. This is likely because the chronic absence of MG23 induces compensatory changes in gene and protein expression related to SR Ca^2+^ regulation [19]. For example, in *Mg23* knockout mice, the expression of calsequestrin 2 and SR Ca^2+^-ATPase 2 was upregulated in fast- and slow-twitch muscles, respectively [19]. These findings are consistent with the physiological properties of fibres lacking MG23: maximal SR Ca^2+^ content was increased in type II fibres, while SR Ca^2+^ uptake was enhanced in type I fibres [19]. Such functional enhancement may attenuate the apparent SR Ca^2+^ leak. In type II fibres, elevated calsequestrin can lower the luminal free-Ca^2+^ concentration, thereby reducing open probability of RyR and decreasing RyR-mediated SR Ca^2+^ leak. In type I fibres, enhanced SR Ca^2+^ uptake allows leaked Ca^2+^ to be efficiently re-sequestered into the SR. The absence of detectable or substantial SR Ca^2+^ leak in skeletal muscle lacking MG23 may result from these compensatory adaptations.

MG23-mediated Ca^2+^ leak in skeletal muscle appears to be accelerated under stressed conditions such as muscle fatigue (Figure 4). Skeletal muscle fatigue is defined as force depression during and after repeated muscle contractions [59]. Multiple factors, including metabolic and mechanical changes, contribute to muscle fatigue [60]. Decreased SR Ca^2+^ release is considered to be a predominant cause of muscle fatigue, especially in the early recovery phase [61,62], and RyR-mediated SR Ca^2+^ leak is reported to be accelerated during the recovery of fatigue [14]. Additionally, other channel(s) might be involved, since RyR inhibitors are unable to completely attenuate fatigue-induced SR Ca^2+^ leak [14]. MG23 likely enhances this SR Ca^2+^ leak, because MG23 knockout mice show faster recovery of fatigue and the fatigue-induced SR Ca^2+^ leak is alleviated in the fibres lacking MG23 (unpublished observation, Watanabe D, Nishi M, Takeshima H). Taken together with minimal effects of MG23 under resting conditions, MG23-mediated SR Ca^2+^ leak may be enhanced under conditions of stress.

Mitochondrial Ca^2+^ buffering is crucial, as it can accelerate recovery of muscle force after damaging eccentric contractions where muscles are stretched while contracting [26]. Although eccentric contractions can typically cause a greater elevation of cytoplasmic Ca^2+^ compared to other contraction types [63], the ability of mitochondrial Ca^2+^ buffering is a significant determinant of fatigue. Interestingly, muscle fibres lacking MG23 show a decrease in Ca^2+^ content in non-SR membranous components during fatigue, while this content remains unchanged in fibres expressing MG23. This suggests that MG23 may enhance mitochondrial Ca^2+^ buffering, either directly or indirectly, under fatigued conditions, though further investigation is required to confirm this mechanism.

The absence of MG23 appears to be beneficial for skeletal muscle fatigue. This raises the important question of what the evolutional benefit is of preserving MG23 expression across multiple species. Transient attenuation of muscle fatigue may exert negative effects on muscle adaptation induced by exercise training. Training-induced adaptations such as improved fatigue resistance are thought to be triggered in part by SR Ca^2+^ leak, because the Ca^2+^ signal can enhance mitochondrial biogenesis [7,64]. MG23-mediated Ca^2+^ leak may therefore play a significant role in muscle adaptation rather than temporal muscle fatigue. However, this possibility has not yet been tested and warrants future investigation.

## 5. Role of Mitsugumin 23 in Cardiac Muscle

In cardiac tissue, exacerbated diastolic SR Ca^2+^ leakage is attributed to progressive deterioration of cardiac function [65,66], implying dysfunction in mechanisms controlling Ca^2+^ flux across the SR membrane [67,68,69]. As the main pathway for Ca^2+^ release from SR stores, it is not surprising that the RyR2 channel has been the primary focus of research investigating the mechanisms of SR Ca^2+^ leak in cardiac muscle, but its relevance as a causative factor rather than phenotypic consequence is questioned [70]. Several non-selective cation channels, including MG23, TRP channels, pannexin channels, and presenilins are found within the SR membrane in cardiac muscle and may function as Ca^2+^-leak channels (for a review, see [34]). According to the protein atlas and further confirmed by blot hybridization analysis of RNA from rabbit tissue [31], MG23 is expressed with particular abundance in cardiac tissue on ER/SR membranes. MG23 also possesses the necessary biophysical properties to contribute to SR Ca^2+^ leak. The role of MG23 as a major player in the leakage of Ca^2+^ across the SR is starting to gain traction.

In heart failure models, RyR2 channels not only display irregular activity and are open when they should remain closed, but the reported mode of channel gating is also altered. In failing human and canine hearts, Marx et al. reported that ∼15% of RyR2 channels are gated in a long-lasting sub-conductance state [65]. Using SR vesicles prepared from both sheep and mouse cardiac tissue, we have shown that at 0 mV and using Ca^2+^ as the permeant ion, the current amplitude of the MG23 full-open state is consistent with that previously reported for RyR2 sub-conductance gating. By incorporating SR vesicles prepared from *Mg23* knockout mice into artificial bilayers under voltage-clamp conditions with Ca^2+^ as the permeant ion and elevating the cytosolic Zn^2+^ to pathophysiological levels, we never observed MG23 or RyR2 sub-conductance state gating, whereas functional full open-state RyR2 channels were evident [20]. This suggests that during cardiac dysfunction, RyR2 does not display sub-conductance gating, but rather, the activity of MG23 is increased and MG23 gating becomes more apparent, suggesting a role for MG23 in the failing heart.

Pathogenic leakage of Ca^2+^ from the SR after pressure overload has been shown to accelerate the development of cardiac hypertrophy and heart failure [71]. The question remains whether altered patterns of Ca^2+^ release and reuptake associated with ECC affect hypertrophic signalling pathways and drive pathology. Using a cardiac pressure overload model, induced by 10-day subcutaneous infusion of Angiotensin II (AngII, 1.1 mg/kg/day) via osmotic pump, our recent unpublished work suggests that the knockout of *Mg23* may protect the heart against left ventricular hypertrophy and altered compliance. We also show that overexpression of MG23 in H9C2 cells correlates with reduced SR Ca^2+^ store levels, linking MG23 to the regulation of Ca^2+^ homeostasis. Interestingly, in the cardiac tissue of MG23 knockout mice, our preliminary results show that other Ca^2+^-handling proteins including RyR2, NCX1, SERCA2, Cav1.2, junctophilin, and calsequestrin are unaltered. We also provide the first direct evidence that MG23 plays a role in the leakage of Ca^2+^ from internal stores in cardiac tissue and that MG23 expression is increased in pathology [21].

Due to the heart’s low regenerative capacity, cardiomyocyte death is a key driver of heart failure and other cardiovascular diseases [72]. There is evidence that MG23 is involved in cell death. In response to endoplasmic reticulum (ER) stress, Yamazaki and co-workers found that MG23 overexpression renders cells more sensitive to etoposide, an anti-neoplastic agent that induces double-stranded DNA breaks [73]. Changes in ER/SR Ca^2+^ content and the leakage of Ca^2+^ from intracellular stores can affect the sensitivity of cells to apoptotic death [74].

Diastolic heart failure may involve many factors other than the regulation of intracellular Ca^2+^. That said, it is of high importance to understand the regulation of diastolic SR Ca^2+^ leak. Novel strategies aimed at normalizing Ca^2+^ leak in heart failure have revealed that the complete elimination of Ca^2+^ fluxes would be catastrophic. Emerging strategies should therefore be focused on modulating rather than blocking Ca^2+^ leak [70]. New therapeutic strategies for heart failure aimed at stabilizing RyR2 channels to fix the leak are now being tested [75,76]. However, if RyR2 is not the major contributor to pathogenic SR Ca^2+^ leakage, then this approach will not completely restore cardiac function. Indeed, in a recent study by Mohamed and co-workers, it was shown that the RyR2-stabilizing agent rycalS36 markedly reduced ventricular arrhythmias but did not attenuate contractile dysfunction [77]. Another recent strategy under active investigation to restore dysregulated myocardial Ca^2+^ homeostasis is the development of small peptide activators of SERCA. Istaroxime is one such peptide. Animal studies and early stage trials suggest that istaroxime decreases the heart rate and shortens the QTC interval [78,79], but improvement in cardiac output was quite modest and only occurred at high doses [80,81]. What is clear is that new strategies to enhance regulation of SR Ca^2+^ dynamics are urgently required.

## 6. Role of Mitsugumin 23 Independent of Ca^2+^ Channel Function

MG23 likely plays a significant role in an apoptotic signalling pathway which originates from the ER. Early work by Yamazaki et al. (2010) reported that MG23 overexpression enhanced apoptosis induced by DNA-damaging drug in the HEK293 cell, while apoptosis was suppressed in MG23 knockout cells [73]. In agreement with this, a later study, also on HEK293 cells, showed that MG23 plays a protective role against UVC-induced DNA damage signals by binding to and accumulating the small heat-shock protein αB-Crystallin in close vicinity of the ER [82]. Recruitment of cytoplasmic proteins to the ER might be a protective mechanism whereby stress-responsive chaperones and scaffold proteins are temporarily sequestered to stabilize the ER’s structure. Consistent with MG23’s role in cellular fate, *TMEM109*, the gene encoding MG23, has been raised as a gene biomarker for pancreatic head cancer [83] and myeloid leukemia [84] by recent studies. Further studies will be required to determine whether this MG23-mediated recruitment constitutes a generalizable stress-response module or if MG23 can also contribute to cell death in a complex in vivo environment.

The signal-recognition particle-dependent (SND) pathway involves the process of importing ER proteins into the ER’s membrane. Using proteomic and quantitative mass spectrometry approaches to assess how protein abundance changes when hSnd2, the human ortholog of the SND protein-targeting pathway component 2, is depleted, Tirincsi and co-workers recently showed that MG23 may be part of the SND pathway, supporting a role for MG23 in ER protein targeting in response to cell stress [85]. This supports the role of MG23 as an ER membrane receptor partner of the SND-targeting pathway for protein trafficking.

In glioblastoma cell lines, ferroptosis is lower compared to normal glial cells. Recent data have shown that the zinc finger and BTB domain-containing protein 20, promote ferroptosis by transcriptionally repressing the expression of *TMEM109* [86]. Given that MG23 plays a role in ER/SR Ca^2+^ responses and its putative role in cell stress signalling, its downregulation may sensitize cells to ferroptotic stress. Modulating this axis could be a therapeutic strategy to help combat glioblastoma progression or therapy resistance. Given the unique ability of MG23 to form a macromolecular complex [32], an important and currently unanswered question is whether the production of MG23 complexes within the SR/ER membrane leads to the destruction of ionic gradients across the ER to drive and/or regulate cell death.

## 7. Future Works

What is clear is that we need to determine the physiological and pathological conditions under which MG23 plays a significant role. If MG23 is revealed to be a promising therapeutic target for the treatment of muscle dysfunction, then further investigation into the biophysical properties of MG23 with the focus on the discovery of innovative drugs to target channel gating is therefore of high importance. Since MG23 is ubiquitously expressed in all tissues, the off-target effects of MG23 modulation as a therapeutic strategy should be considered. Given that MG23 is more abundantly expressed in muscle compared to other tissues, it is likely that muscle tissue will be more impacted by any changes in MG23’s behaviour. Given the potential role of MG23 as a contributor and/or regulator of basal cellular Ca^2+^ leak, it could be proposed that dysregulated MG23 signalling drives early stages of remodelling in muscle disorders. As the basal Ca^2+^ leak becomes unregulated, as a direct result of increased MG23 channel activity or increased MG23 expression, this could drive the cell down a pathogenic pathway by recruiting and activating other major Ca^2+^-release channels, including RyR and IP3R. This would contribute further to the pathogenic Ca^2+^ leak pathway. How MG23 crosstalks with RyR2 and IP3R is an underexplored but important area of research. Further investigation into the formation of the oligomeric structures of MG23 and how these are linked to cellular function also warrants further research as complex formation is reported to influence MG23-mediated SR Ca^2+^ leak. Identifying factors that enhance complex formation may represent a promising future direction for therapeutic potential. If MG23 is revealed to be a new target for the treatment of pathologies linked to the leakage of Ca^2+^ from the SR, then a high-resolution structure of MG23 is required to translate basic science findings into a structure-based drug-design programme. The manipulation of MG23 function may help tackle pathologies driven by the dysregulated leak of Ca^2+^ from SR stores, including heart failure and muscle fatigue. Uncovering pharmacological compounds and having a tool kit able to regulate MG23 will also aid in our understanding of the role of MG23 in other conditions where the leakage of Ca^2+^ from the ER/SR is thought to play a key role.

SR Ca^2+^ leak is currently recognized as a key mechanism underlying many types of muscle diseases. This leak is regulated by various channels, including RyR and MG23. Current research predominantly investigates RyR. However, targeting RyR represents a potential disadvantage: RyR inhibition may compromise normal SR Ca^2+^ release, because RyR acts as an SR Ca^2+^-release channel as well as a Ca^2+^-leak channel. Conversely, MG23 does not appear to influence SR Ca^2+^ release [19], suggesting that selectively targeting MG23 offers a more advantageous strategy to mitigate SR Ca^2+^ leak. Moreover, the absence of reported negative effects from either MG23 knockdown or knockout on any tissues further supports this approach. While the combined inhibition of both RyR and MG23 could lead to greater attenuation of the SR Ca^2+^ leak, this is likely suboptimal, as excessive reduction in SR Ca^2+^ leak negatively impacts cellular function and adaptation.

## 8. Summary

MG23 likely acts as SR Ca^2+^-leak channel and may play a significant role in muscle adaptation under pathophysiological conditions such as skeletal muscle fatigue and heart failure, where SR Ca^2+^ leak is a key mechanism. In the future, identifying specific inhibitors of MG23 will be an important step toward potential therapeutic applications in humans.

## Figures and Tables

**Figure 1 cells-14-01958-f001:**
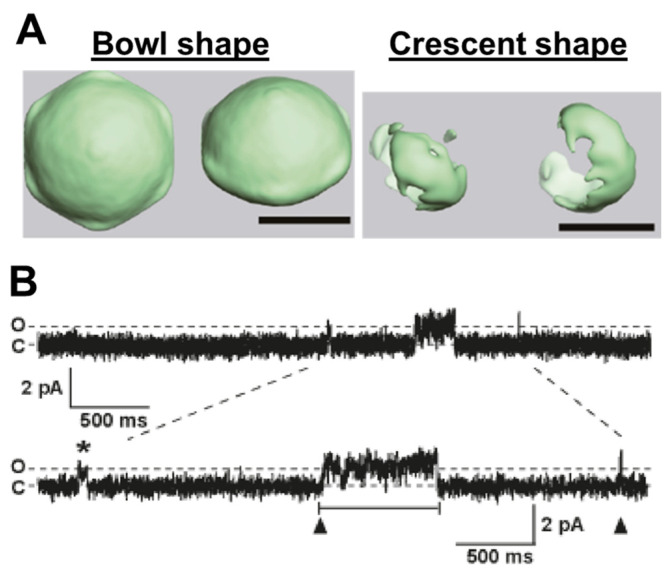
Properties of Mitsugumin 23 (MG23) channel. (**A**) Three-dimensional reconstruction of electron miscopy images for native MG23. The scale bars are 10 nm. (**B**) Example of MG23 channel gating. C indicates all channels closed; O indicates single MG23 opening. Both single (asterisk) and coordinated (arrowhead) gating of MG23 channel occurred. Adapted from Venturi et al. (2011) [32].

**Figure 2 cells-14-01958-f002:**
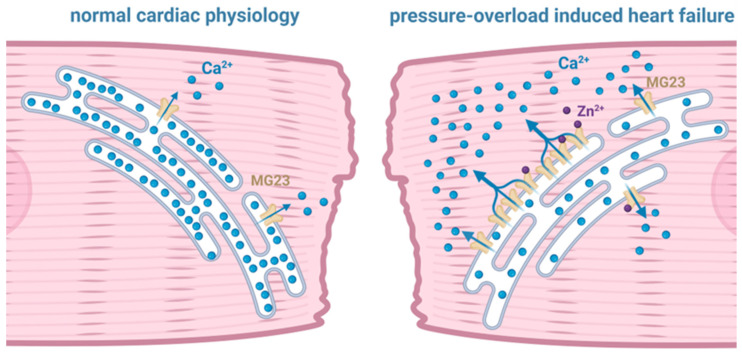
Illustration of the potential contribution of Mitsugumin 23 (MG23) to intracellular Ca^2+^ dynamics that may drive cardiac muscle dysfunction. As shown by the blue arrows, the sarcoplasmic reticulum in pressure-overload induced heart failure appears to leak Ca^2+^ through MG23.

**Figure 3 cells-14-01958-f003:**
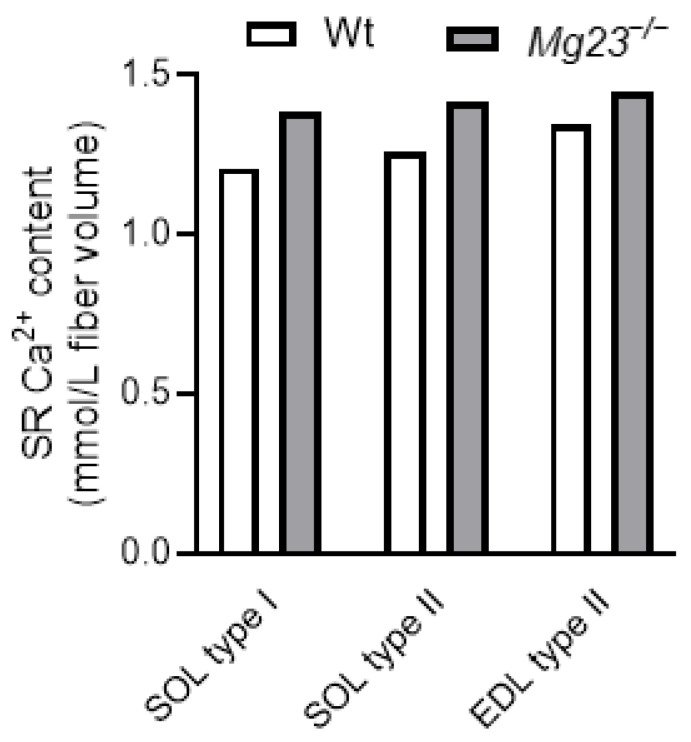
Sarcoplasmic reticulum (SR) Ca^2+^ content in skeletal muscle fibres of Mitsugumin 23 knockout (*Mg23*^−/−^) mice. EDL, extensor digitorum longus; SOL, soleus; Wt, wild type. Adapted from Watanabe et al. (2024) [19].

**Figure 4 cells-14-01958-f004:**
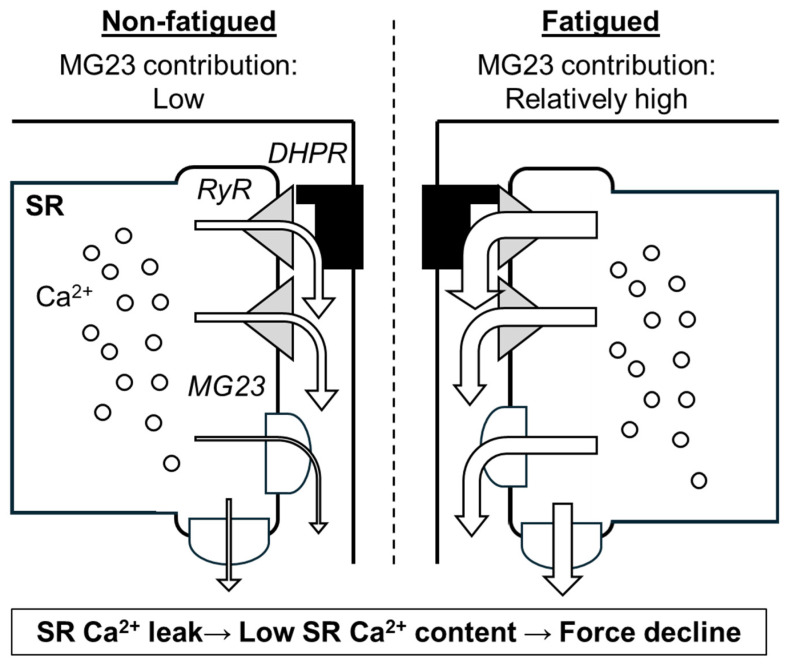
Scheme of predicted contribution of Mitsugumin 23 (MG23) to skeletal muscle fatigue. As shown by arrows, sarcoplasmic reticulum (SR) Ca^2+^ leak through ryanodine receptor (RyR) and MG23 is thought to occur even under resting condition. During fatigue, RyR-mediated and MG23-mediated Ca^2+^ leak are enhanced. DHPR, dihydropyridine receptor.

## Data Availability

No new data were created or analyzed in this study.

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
