# Peer review of "Predicted Role of Mitsugumin 23 in Skeletal and Cardiac Muscle"

_cells, 2025, doi:10.3390/cells14241958_

Round 1
Reviewer 1 Report
Comments and Suggestions for Authors
This review by Watanabe et al. discusses the role of Mitsugumin 23 (MG23/TMEM109), a transmembrane cation channel expressed in the ER/SR, in skeletal and cardiac muscles. The authors summarise evidence supporting MG23 as a SR Ca2+-leak channel and contributing to calcium handling. They explore its implications in muscle fatigue and heart failure, and highlights future therapeutic prospects. The manuscript is well-written and scientifically sound. It addresses an emerging topic of substantial interest in muscle physiology and ion channel biology. Figures are concise and informative. Overall, this is a valuable contribution to the field. However, several improvements would enhance the quality of the review:
Major Comments
- What are the potential regulatory mechanisms that could modulate MG23 channel activity (e.g. phosphorylation, redox state, lipid environment)?
- Provide more detail on MG23 non-selective cation channel properties. Specifically, discuss its permeability to Ca2+ and other physiological ions such as K+, Na+ and Zn2+, and if possible provide the relevant permeability ratios.
- Discuss the paper from Yamashita et al. (https://pubmed.ncbi.nlm.nih.gov/23542032/) showed a protective role for MG23. Specifically, knockdown of MG23 enhances cell death induced by UV.
- Discuss the effects of MG23 overexpression (a gain-of-function model) to complement the discussion of loss-of-function models.
- What is the relative quantitative contribution of MG23 versus RyR2 (and all other discussed channels) to the total SR Ca2+-leak flux in both skeletal and cardiac muscle under basal and pathological conditions?
- Discuss the therapeutic potential of combination strategies that target multiple leak channels (MG23, RyR2, TRP channels, etc.) to enhance clinical efficacy.
- Discuss potential side effects or tissue specificity concerns of targeting MG23.
Minor Comments
- Ensure correct subscript formatting for ions Ca2+ vs Ca2+
- Correct minor typographical errors (e.g. “reconstraction”)
- Increase resolution of Figures 1-2
- For Figure 2, provide individual data points and appropriate statistical analyses, to determine whether the differences shown are statistically significant.
Author Response
Thank you for your valuable comments. We have now added discussions you pointed out including potential regulators of MG23 and the contribution of MG23 to cell death. We have responded sequentially to your comments.
Major comment 1: What are the potential regulatory mechanisms that could modulate MG23 channel activity (e.g. phosphorylation, redox state, lipid environment)?
Response: We have added a new section discussing potential regulators for MG23 activity entitled “Potential regulators”. In this section, zinc, oxidative stress and phosphorylation have been discussed.
Major comment 2: Provide more detail on MG23 non-selective cation channel properties. Specifically, discuss its permeability to Ca2+ and other physiological ions such as K+, Na+ and Zn2+, and if possible provide the relevant permeability ratios.
Response: We have now provided further detail regarding the relative permeability of MG23 as follows: “The relative permeability of MG23 to Ca2+ and K+ is reported to be equal. Using the Fatt-Ginsborg equation to calculate the relative permeability of MG23 following construction of current-voltage relationships under bi-ionic conditions the PCa2+/PK+ was reported to be 1:1. Given the location of MG23 and the large Ca2+ gradient across the SR membrane, it was suggested that MG23 may play a role alongside RyR2 in shaping intracellular Ca2+-dynamics [20, 32]. The permeability of MG23 to other monovalent and divalent cations has not been reported.”.
Major comment 3: Discuss the paper from Yamashita et al. (https://pubmed.ncbi.nlm.nih.gov/23542032/) showed a protective role for MG23. Specifically, knockdown of MG23 enhances cell death induced by UV.
Response: We have discussed the paper by Yamashita et al. in the new section entitled “Role of Mitsugumin 23 independent of Ca2+ channel function”.
Major comment 4: Discuss the effects of MG23 overexpression (a gain-of-function model) to complement the discussion of loss-of-function models.
Response: Thank you for highlighting this. We have now included work by Yamashita and also the work by Titinsci and Chen. All of these papers link MG23 to regulation of ER stress and apoptotic pathways and discuss gain of function models.
Major comment 5: What is the relative quantitative contribution of MG23 versus RyR2 (and all other discussed channels) to the total SR Ca2+-leak flux in both skeletal and cardiac muscle under basal and pathological conditions?
Response: The relative contribution of MG23 to other channels remains unclear, and we believe that the relative contribution fluctuates depending on the situation. We have added comments regarding this as follows: “In heart failure models, RyR2 channels not only display irregular activity and are open when they should remain closed, but the reported mode of channel gating is also altered. In failing human and canine hearts, Marx et al. reported that ∼15% of RyR2 channels gated in a long-lasting sub-conductance state [65]. Using SR vesicles prepared from both sheep and mouse cardiac tissue, we have shown that at 0 mV and using Ca2+ as the permeant ion, the current amplitude of the MG23 full-open state is consistent with that previously reported for RyR2 sub-conductance gating. By incorporating SR vesicles prepared from Mg23 knock-out mice into artificial bilayers under voltage-clamp conditions with Ca2+ as permeant ion and elevating the cytosolic Zn2+ to pathophysiological levels, we never observed MG23 or RyR2 sub-conductance state gating, whereas functional full open state RyR2 channels were evident [20]. This suggests that in cardiac dysfunction RyR2 does not display sub-conductance gating, but rather, the activity of MG23 is increased and MG23-gating becomes more apparent, suggesting a role for MG23 in the failing heart.”.
Major comment 6: Discuss the therapeutic potential of combination strategies that target multiple leak channels (MG23, RyR2, TRP channels, etc.) to enhance clinical efficacy.
Response: We have discussed the combined treatments in the section “Future works” as follows: “SR Ca2+-leak is currently recognized as a key mechanism underlying many types of muscle diseases. This leak is regulated by various channels, including RyR and MG23. Current research predominantly investigates RyR. However, targeting RyR represents a potential disadvantage: RyR inhibition may compromise normal SR Ca2+ release, because RyR acts as SR Ca2+-release channel as well as Ca2+-leak channel. Conversely, MG23 does not appear to influence SR Ca2+-release [19], suggesting that selectively targeting MG23 offers a more advantageous strategy to mitigate SR Ca2+-leak. Moreover, the absence of reported negative effects from either MG23 knockdown or knockout on any tissues further supports this approach. While the combined inhibition of both RyR and MG23 could lead to greater attenuation of the SR Ca2+-leak, this is likely suboptimal as excessive reduction of SR Ca2+-leak negatively impacts on cellular function and adaptation.”.
Major comment 7: Discuss potential side effects or tissue specificity concerns of targeting MG23.
Response: We have discussed the side effects as well as combined treatments in the section “Future works” (pleases see response for your previous comment).
Minor comment 1: Ensure correct subscript formatting for ions Ca2+ vs Ca2+
Minor comment 2: Correct minor typographical errors (e.g. “reconstraction”)
Response: We have amended typos in the revised manuscript. Actually, our original manuscript does not have these typos, but the Journal converted our manuscript to the journal’s format. These typos occurred by the process of the conversion. We have now submitted the revised ms using journal format.
Minor comment 3: Increase resolution of Figures 1-2
Response: We have increased the resolution of Fig. 1 and current Fig. 3. Similar to the typographical issues, the lower resolution was likely caused by the journal's file conversion process.
Minor comment 4: For Figure 2, provide individual data points and appropriate statistical analyses, to determine whether the differences shown are statistically significant.
Response: We have kept the current form because the SR Ca2+ content is expressed only as average difference between total Ca2+ content in membrane and non-SR Ca2+ content [please see detail in Watanabe et al. (2024)]. Nonetheless, statistical tests were applied to total Ca2+ content in membranous components and non-SR Ca2+ content.
Reviewer 2 Report
Comments and Suggestions for Authors
In the review, the authors clearly summarized the expression, structure and function of MG23 as a calcium “leak” channel on the SR membrane of skeletal and cardiac muscle. Overall, this review is concise and straightforward in highlighting the role of MG23 in calcium leak that contributes to skeletal muscle fatigue and heart failure.
There are a few minor issues:
- There are some discrepancies in the formats of the text and references, such as different fonts in page 1 and 3 for the texts; the styles of “Ca2+” need to be unified as well.
- Consider adding zinc homeostasis into the “channel property” section as it is known as a main regulator of MG23 gating and was mentioned multiple times in this review. Moreover, the fluctuation of zinc level might give rise to the enhanced calcium leak in skeletal muscle as indicated in Figure 3.
- Please specify the pathological condition in Figure 4 (pressure overload or diastolic heart failure) and change “pathological cardiac physiology” to match that specific condition.
- In the discussion on regulation of MG23 by zinc, please suggest how the zinc level changes in pressure overload and/or diastolic heart failure to make this part more relevant.
Author Response
Thank you for your valuable comments. In the revised manuscript, we have moved the statement of zinc to the front and discussed more. We have responded sequentially to your comments.
Comment 1: There are some discrepancies in the formats of the text and references, such as different fonts in page 1 and 3 for the texts; the styles of “Ca2+” need to be unified as well.
Response: We have amended all typos in the revised manuscript. This is likely due to the format change by the Journal.
Comment 2: Consider adding zinc homeostasis into the “channel property” section as it is known as a main regulator of MG23 gating and was mentioned multiple times in this review. Moreover, the fluctuation of zinc level might give rise to the enhanced calcium leak in skeletal muscle as indicated in Figure 3.
Response: We have now a created new section entitled “Potential regulators”. In skeletal muscle, zinc contribution remains uncertain. However we have commented on the potential effects, but we mentioned it in the text, while we did not further describe this in the figure, as at present this is speculative.
Comment 3: Please specify the pathological condition in Figure 4 (pressure overload or diastolic heart failure) and change “pathological cardiac physiology” to match that specific condition.
Response: We have now amended previous Figure 4.
Comment 4: In the discussion on regulation of MG23 by zinc, please suggest how the zinc level changes in pressure overload and/or diastolic heart failure to make this part more relevant.
Response: We have now added a section to discuss how changes in cellular Zn2+ levels may regulate MG23 to increase SR Ca2+ leak and drive pathology as follows: “We do know that gating of MG23 prepared from cardiac tissue is regulated by Zn2+ [20] (Fig. 2). It is likely that MG23 in skeletal muscle is also regulated by Zn2+ as it is coded by the same gene and to date there is no evidence to suggest that MG23 is expressed as a different isoform in different tissue. Cardiomyocytes contain a small but measurable pool of free Zn2+ in the cytosol, which is reported to be ∼100 pM [35]. This basal level of Zn2+ is transiently altered during cardiac ECC as result of both the influx of Zn2+ into the cell through L-type Ca2+ channels and the release of Zn2+ from intracellular stores including the ER/SR [36]. Disrupted intracellular Zn2+-homeostasis has long been associated with reduced cardiac contractility observed in heart failure [37-39]. Increasing the Zn2+-concentration gradient across the plasma membrane by application of high levels (10 μM to 20 mM) of extracellular Zn2+ was demonstrated to reduce isolated cardiomyocyte contractility as a result of SR-Ca2+ store depletion, suggesting an intimate relationship between raised intracellular Zn2+ and SR Ca2+-release [40, 41]. Indeed, studies suggest that alterations in intracellular Ca2+-homeostasis and intracellular Zn2+ levels are more than coincidental and that they may exist in a synergistic relationship [36, 42]. Using live cell imaging approaches, elevations in intracellular Zn2+ were demonstrated to be in close proximity to the SR [39]. Zn2+-modulation of SR Ca2+-load may therefore play an important role in cardiac contractility. Turan and colleagues have extensively investigated the proposed relationship between elevated intracellular Zn2+-levels, SR Ca2+-release and reduced cardiac contractility. They showed that redox-dependent signalling pathways can elevate intracellular Zn2+ levels by the release of Zn²⁺ from SR stores, and oxidation-induced release of Zn²⁺ from Zn²⁺-binding proteins such as metallothioneins. These mechanisms result in a 30-fold elevation in the cytoplasmic Zn2+ level, resulting in myocardial damage [43-46].
Using cardiac SR vesicles isolated from sheep to study single channel properties under voltage-clamp conditions, our previous work demonstrated that elevating the cytosolic [Zn2+] from 100 pM to 1 nM regulates the gating of both RyR2 and MG23 [20, 47]. Furthermore, marked alterations in the expression of Zn²⁺ transporters have been observed as part of a compensatory response to rising intracellular Zn²⁺ levels in a pressure-overload (TAC) rat model, H9C2 cellular doxorubicin-induced heart-failure model, and in tissue from human failing hearts [21, 48, 49]. Elevated Zn²⁺ concentrations in these settings correlate closely with increased ER-stress markers, indicating that dysregulated zinc homeostasis contributes directly to ER-stress signalling during cardiac remodelling and the failing heart [50]. This suggests that Ca2+-homeostasis is intimately related to intracellular Zn2+ levels and suggests that physiological levels of Zn2+ are essential in fine-tuning the release of Ca2+ from the SR during cardiac ECC. Zn2+ induced SR Ca2+ leak through modulation of both MG23 and RyR2 function may therefore be a key mechanism in the progression of cardiac dysfunction in the failing heart.”.
Reviewer 3 Report
Comments and Suggestions for Authors
This is a comprehensive review of the sate of current knowledge of Mitsugumin 23/ TMEM109 and should be of interest in raising awareness of a role for this molecule in muscle EC coupling.
Comments:
1 Please make more clear that TMEM109 is a synonym for MG23 and include any research on TMEM109 in your analysis
2 Please provide Uniprot/NLM references and preferably a domain map and sequence (locating mutations, if any are known).
3 Can you also determine how many molecules are likely to make up the bowl-shaped objects observed in em. Does the channel activity require oligomerisation?
4 Although the case that MG23 plays a role in EC coupling modulation is reasonably made, you should not describe it as "important". If it were actually important its role would be far more obvious than it actually is. "significant" may be an acceptable alternative.
Author Response
Thank you for your valuable comments. We have responded sequentially to your comments.
Comment 1: Please make more clear that TMEM109 is a synonym for MG23 and include any research on TMEM109 in your analysis
Response: We have stated that TMEM109 is a gene coding MG23 as follows: “Mitsugumin 23 (UniProt reference: Q9BVC6 for human and Q3UBX0 for mouse), encoded by TMEM109 gene, is named by Japanese researchers based on its location within the muscle cell.”.
Comment 2: Please provide Uniprot/NLM references and preferably a domain map and sequence (locating mutations, if any are known).
Response: We now show UniProt reference (please see your previous comment).
Comment 3: Can you also determine how many molecules are likely to make up the bowl-shaped objects observed in em. Does the channel activity require oligomerisation?
Response: We have suggested the predicted molecular number required for crescent- and bowl-shaped molecules in the text and commented that oligomerisation is likely to contribute to channel activity as follows: “The exact number of MG23 subunits required for the formation of crescent- and bowl-shaped molecules remains uncertain. However, based on simulations derived from electron microscopy images, it has been proposed that the crescent-shaped molecule is composed of a hexamer of MG23, and the bowl-shaped molecule is assembled from six crescent-shaped molecules [32].”.
Comment 4: Although the case that MG23 plays a role in EC coupling modulation is reasonably made, you should not describe it as "important". If it were actually important its role would be far more obvious than it actually is. "significant" may be an acceptable alternative.
Response: We have amended text as you suggested.
Round 2
Reviewer 1 Report
Comments and Suggestions for Authors
Authors have addressed all my comments. This is a great review.